# PeerJ

# Obesity increases operating room times in patients undergoing primary hip arthroplasty: a retrospective cohort analysis

Bassam Kadry[1], Christopher D. Press[1], Hassan Alosh[2], Isaac M. Opper[3], Joe Orsini[3], Igor A. Popov[3], Jay B. Brodsky[1] and Alex Macario[1]

[1] Department of Anesthesiology, Perioperative and Pain Medicine, Stanford University Medical Center, Stanford, CA, USA
[2] Department of Orthopaedic Surgery, Hospital of the University of Pennsylvania, Silverstein, Philadelphia, PA, USA
[3] Stanford University Economics Department, Stanford, CA, USA

Corresponding author
Jay B. Brodsky,
Jbrodsky@stanford.edu

## ABSTRACT

**Background.** Obesity impacts utilization of healthcare resources. The goal of this study was to measure the relationship between increasing body mass index (BMI) in patients undergoing total hip arthroplasty (THA) with different components of operating room (OR) time.

**Methods.** The Stanford Translational Research Integrated Database Environment (STRIDE) was utilized to identify all ASA PS 2 or 3 patients who underwent primary THA at Stanford Medical Center from February 1, 2008 through January 1, 2013. Patients were divided into five groups based on the BMI weight classification. Regression analysis was used to quantify relationships between BMI and the different components of total OR time.

**Results.** 1,332 patients were included in the study. There were no statistically significant differences in age, gender, height, and ASA PS classification between the BMI groups. Normal-weight patients had a total OR time of 138.9 min compared 167.9 min ($P < 0.001$) for morbidly obese patients. At a BMI $> 35$ kg/m$^2$ each incremental BMI unit increase was associated with greater incremental total OR time increases.

**Conclusion.** Morbidly obese patients required significantly more total OR time than normal-weight patients undergoing a THA procedure. This increase in time is relevant when scheduling obese patients for surgery and has an important impact on health resource utilization.

## INTRODUCTION

Using the World Health Organization's (WHO) weight-class definitions, two-thirds of the adult population in the United States above the age of 60 years are "pre-obese" or "obese" (*WHO, 2000*; *Flegal et al., 2010*). Furthermore, the prevalence of obesity is projected to

increase by >33% over the next twenty years (*Finkelstein et al., 2012*). The current obesity "epidemic" has had a great impact on healthcare resources, and its effects will increase further as the incidence of obesity continues to rise (*Wolf & Colditz, 1998*; *Allison, Zannolli & Narayan, 1999*). In addition to an increased risk of postoperative complications, hospital readmission rates are increased for obese patients (*Farkas et al., 2012*; *Reinke et al., 2012*). There is evidence that the duration of certain surgical procedures and the total amount of time spent in the operating room (OR) are also increased for obese patients (*Batsis et al., 2009*; *Wang et al., 2013*). This is relevant since longer OR times can be extrapolated to an increase in resource expenditures. Previous research has been conducted looking at the increases in direct cost and length of stay in total hip arthroplasty (*Kremers et al., 2014*). For example, if not scheduled accurately, an operation running later than planned will increase the potential need for overtime help (*Hirose et al., 2011*). The increased costs associated with the obese surgical patient are not accompanied by similar increases in third party payments thereby placing financial risk on facilities that care for these patients (*Silber et al., 2012*).

The goal of this study was to measure the relationship between increasing obesity in patients undergoing total hip arthroplasty (THA) with the various components of OR time in order to determine how obesity contributes to overall case duration. The expected result is an increase in resource utilization for obese surgical patients as measure by perioperative times.

## MATERIALS AND METHODS

Following IRB approval, the Stanford Translational Research Integrated Database Environment (STRIDE) was utilized to identify all patients who underwent primary THA at Stanford Medical Center from February 1, 2008 through January 1, 2013. STRIDE is a standards-based informatics platform supporting clinical and translational research and includes a clinical data warehouse based on the HL7 Reference Information Model (*Lowe et al., 2009*). This observational research study utilized the STROBE guidelines for improved reporting of the findings (*von Elm et al., 2008*).

Only American Society of Anesthesiologists Physical Status (ASA PS) class 2 or 3 patients undergoing primary elective unilateral THA procedures under general anesthesia, with or without spinal for post-operative pain control were included in the study. ASA 1, 4 and 5 patients were removed from the study since they represent a small number of the total hip arthroplasties. Patients were divided into five groups based on WHO BMI classification (*WHO, 2000*). Group 1—normal-weight (BMI = 18.5–24.9 kg/m$^2$), Group 2—pre-obese (BMI = 25–29.9 kg/m$^2$), Group 3—obese (BMI—30–34.9 kg/m$^2$), and Group 4—severely obese (BMI = 35–39.9 kg/m$^2$) and Group 5—morbidly obese (BMI >= 40.0 kg/m$^2$).

The following parameters were retrieved: age, gender, height, weight, surgeon (14 surgeons), anesthesiologist (116 anesthesiologists) and ASA PS. Patients who had placement of central lines or arterial lines were excluded from the study since these interventions could be a confounding variable for increased length of perioperative time.

Time stamps were retrieved from the STRIDE database. The data points collected were time of (a) Admission to pre-op Unit, (b) Entry into operating room, (c) Anesthesia handoff, (d) Surgery start, (e) Surgery end, (f) Out of operating room, (g) Post-Anesthesia Care Unit (PACU) admission, (h) PACU discharge, and (i) Hospital discharge day and time.

The primary outcome variable of the study was total OR Time, which is the time interval beginning when a patient entered the OR (b) until the moment the patient physically left the OR (f).

Secondary outcome variables included:

1. Induction Time defined as the time from Entry into OR (b) to Anesthesia Handoff (c). (Anesthesia handoff is the time the anesthesiologist completed his/her work and turned the patient over to the surgical team for positioning, preparation, and draping.)
2. Operation time defined as the interval from Surgery Start (d) to Surgery End (e).
3. Emergence Time defined as the interval from Surgery End (e) to Out of Operating Room (f).
4. Recovery Room Time defined as the interval from PACU Admission (g) to PACU Discharge (h).
5. Total Hospital Length of Stay defined as the interval from the Admission to Pre-op Unit (a) to Hospital Discharge (i).

All time data points were collected and recorded into the electronic medical record by pre-operative, operating, and recovery room nurses who were unaware of the study, as data was analyzed retrospectively.

## STATISTICAL ANALYSIS

Observations taken from the STRIDE database, which already excluded patients based on ASA, central line and/or arterial line placement, and regional anesthesia, resulted in 1,613 patients. The top 1% and bottom 1% times for each data interval were excluded as a means of removing incorrectly entered data points and ensuring outliers did not drive the results (185 patients removed). Patients with a BMI of greater than 50 kg/m$^2$ or less than 18.5 kg/m$^2$ were removed from the study due to a paucity of data points (96 patients removed). These two procedures would cause a wide variability in induction times and could be representative of inability to get a non-invasive blood pressure or peripheral intravenous access or could be due to severity of illness. By removing these few occurrences from our dataset, we reduce the possibility of variability due to covariates and had a total of 1,332 patients remaining. All data points for the time intervals were available for these patients, except 1 was missing for induction time (1,331 observations), 4 were missing for operation time (1,328 observations), and 568 were missing for recovery time (764 observations). The patients were grouped by body mass index as defined by the WHO classifications (stated above in this section). For the primary and each secondary outcome variable the mean and standard deviations were calculated. We estimated regression parameters separately for each time component of the case. In the data analysis BMI is

an indicator variable. The database accessible covariates that can affect case duration were included as regressors (e.g., patient age, gender, height, and ASA PS classification) (Table 5). These were included to remove any variability in demographics, size, and severity of illness between our patient's in their BMI groupings. Multiple *t*-test assuming unequal variance were performed to compare the Group 1 (normal weight) BMI patients to each successively higher BMI grouping. This analysis was repeated for the primary and each secondary outcome variables.

A graphical regression analysis was performed to quantify the relationship between BMI (kg/m$^2$) and total operating room time, induction time, total operation time, emergence time, recovery room time, and length of hospital stay. Using Stata (StataCorp LP, College Station, Texas) a kernel weighted local polynomial regression to examine potential non-linearity in the relationship between BMI, the continuous variable, and procedure duration was performed. In the graphical analysis the covariates are not controlled for. This method traces out average total OR time and induction time for each BMI value without making any parametric assumptions.

## RESULTS

A total of 1,332 patients were included in this study. There were no statistically significant differences in age, gender, height, and ASA PS classification between the different study groups Table 1.

Group 1 (normal-weight) patients undergoing THA had a mean Total OR Time of 138.9 min, Group 3 (obese) patients 146.3 min, and Group 5 (morbidly obese) patients 167.9 min (Table 2, Fig. 1). The difference in mean times with adjusted and unadjusted values are presented in Tables 3 and 4. Controlled values are presented in Table 5.

There was also a direct association between increasing BMI and the length of many of the individual components of total OR Time based on *t*-test between higher BMI groups when compared to Normal BMI group. Incremental BMI group increase was associated with a greater incremental time increase at higher BMIs.

Patients in Group 1 had a mean Induction Time of 21.4 min, Group 3 patients 23.7 min, while Group 5 patients required 28.2 min (Fig. 2). Group 5 patients had a longer mean Operation Time of 103.0 min versus 86.1 min for Group 1 patients (Fig. 3). For Group 1 patients, the mean Emergence Time equaled 9.1 min compared to 10.9 min for Group 5. There were no statistically significant differences in mean Recovery Time between groups. Group 5 patients had a significant increased mean Length of Hospital Stay, 3.9 days compared to 3.5 days for Group 1 patients (Table 2).

## DISCUSSION

As the number of obese patients continues to rise so does the demand for joint arthroplasty procedures. Therefore, elucidating the impact of obesity on perioperative costs is becoming ever more relevant (*Perka et al., 2000*; *Kurtz et al., 2007*). Our study of 1,332 patients undergoing primary THA procedures at an academic medical center found that obesity predictably lengthens both induction and surgical times. The induction time is greater as expected due to longer times in moving, positioning, and pre-oxygenating obese patients.

**Table 1 Demographic variables.** Associated standard deviation (when appropriate) and *p* value comparing normal BMI to other BMI groupings.

| BMI range | 18.5–24 | | | 25–29 | | | 30–34 | | | 35–39 | | | 40–50 | | |
|---|---|---|---|---|---|---|---|---|---|---|---|---|---|---|---|
| Variables | | SD | *p* | | SD | *p* | | SD | *p* | | SD | *p* | | SD | *p* |
| Age (mean) | 62.92 | 14.50 | – | 63.77 | 11.93 | 0.53 | 62.20 | 12.13 | 0.48 | 61.29 | 9.80 | 0.45 | 59.02 | 9.54 | 0.39 |
| ASA 2 total (percent) | 175 (49%) | – | – | 241 (48%) | – | 0.53 | 122 (45%) | – | 0.55 | 48 (43%) | – | 0.56 | 29 (44%) | – | 0.56 |
| Male total (percent) | 115 (32%) | – | – | 281 (56%) | – | 0.69 | 147 (54%) | – | 0.68 | 49 (44%) | – | 0.60 | 23 (35%) | – | 0.52 |
| Height in meters (mean) | 1.69 | 0.11 | – | 1.72 | 0.11 | 0.62 | 1.72 | 0.10 | 0.62 | 1.70 | 0.10 | 0.57 | 1.68 | 0.10 | 0.50 |
| Number of patients | 357 | – | – | 502 | – | – | 270 | – | – | 111 | – | – | 66 | – | – |

**Table 2 Outcomes table.** Mean-listed in minutes except total hospital time, standard deviation (SD), $p$-value: heteroscedastic $t$-test results comparing interval times of normal BMI to other BMI groupings.

| BMI range | 18.5–24.9 | | | 25–29.9 | | | 30–34.9 | | | 35–39.9 | | | 40–50 | | | Total number |
|---|---|---|---|---|---|---|---|---|---|---|---|---|---|---|---|---|
| Variables | Mean | SD | p | Mean | SD | p | Mean | SD | p | Mean | SD | p | Mean | SD | p | |
| Total time in OR | 138.9 | 1.4 | – | 141.7 | 1.2 | 0.152 | 146.3 | 1.8 | <0.001 | 156.3 | 2.9 | <0.0005 | 167.9 | 3.8 | <0.0005 | 1,332 |
| Induction time | 21.4 | 0.4 | – | 21.8 | 0.4 | 0.475 | 23.7 | 0.5 | <0.001 | 26.3 | 0.8 | <0.0005 | 28.2 | 1.1 | <0.0005 | 1,331 |
| Operation time | 86.1 | 1.2 | – | 87.4 | 0.9 | 0.425 | 90.3 | 1.4 | 0.022 | 97.6 | 2.4 | <0.0005 | 103.0 | 2.9 | <0.0005 | 1,328 |
| Emergence time | 9.1 | 0.3 | – | 9.1 | 0.2 | 0.921 | 9.2 | 0.3 | 0.784 | 9.5 | 0.4 | 0.452 | 10.9 | 0.7 | 0.006 | 1,332 |
| Recovery time | 184.8 | 3.9 | – | 174.8 | 3.2 | 0.136 | 167.7 | 4.5 | 0.031 | 184.9 | 7.9 | 0.994 | 174.6 | 8.0 | 0.474 | 764 |
| Total hospital stay (days) | 3.5 | 0.1 | – | 3.5 | 0.1 | 0.801 | 3.5 | 0.1 | 0.949 | 3.6 | 0.1 | 0.233 | 3.9 | 0.1 | 0.001 | 1,332 |
| Number of patients | 362 | – | – | 512 | – | – | 275 | – | – | 113 | – | – | 70 | – | – | 1,332 |

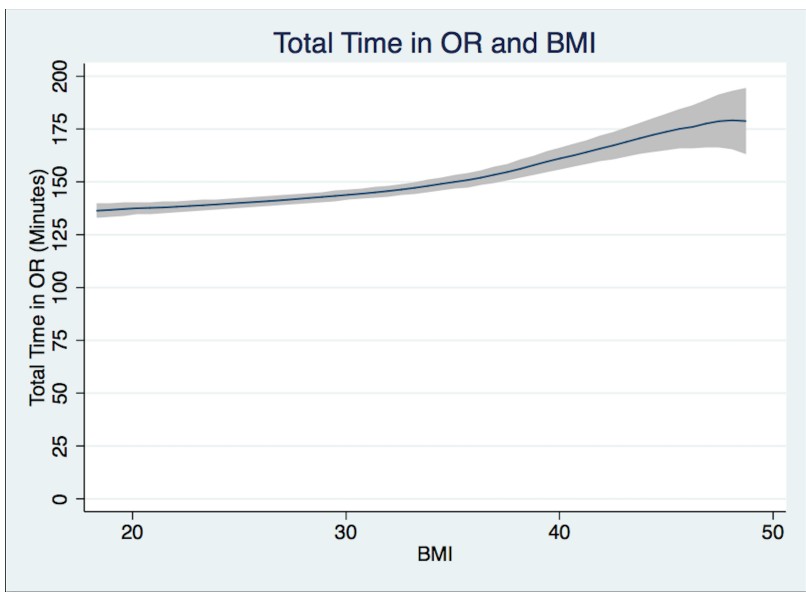

**Figure 1 Total case time.** Total case time (patient in room to patient out of room in minutes) and BMI Polynomial Regression Line (blue line) with 95% CI (gray); (Epanechnikov kernel of degree 0, Bandwidth of 3.24, pwidth of 4.86). The differences in the bandwidth are due to differences in the variance of outcome variables.

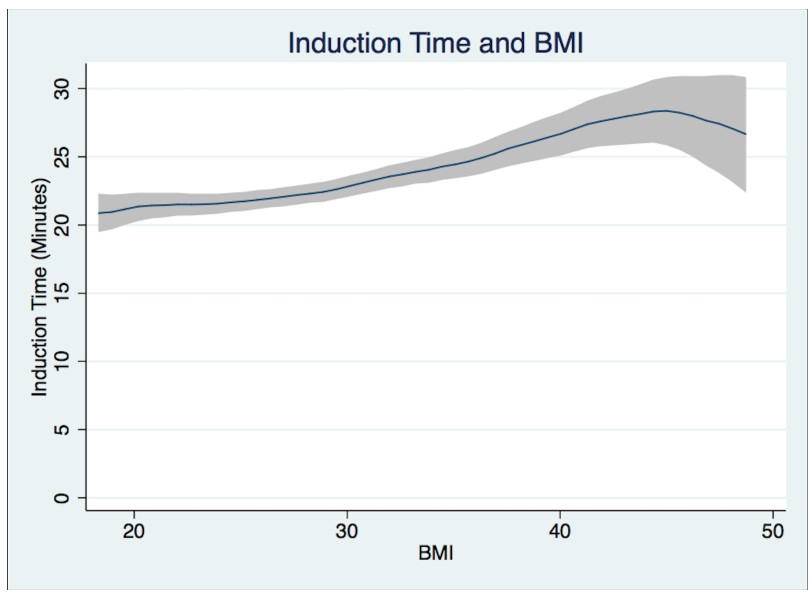

**Figure 2 Induction time.** Induction time (from patient in room to handoff from anesthesiologist to surgeon in minutes) and BMI Polynomial Regression Line (blue line) with 95% CI (gray); (Epanechnikov kernel of degree 0, Bandwidth of 2.2, pwidth of 3.3). The differences in the bandwidth are due to differences in the variance of outcome variables.

**Table 3 Adjusted difference.** Difference in mean times when compared to normal BMI patients (BMI 18.5–24.9), adjusted.

| BMI range | 25–29.9 | | | 30–34.9 | | | 35–39.9 | | | 40–50 | | | Total number |
|---|---|---|---|---|---|---|---|---|---|---|---|---|---|
| | Difference | 95% CI | p | Difference | SD | p | Difference | SD | p | Difference | SD | p | |
| Total time in OR | 3.48 | ±0.170 | 0.077 | 7.49 | ±0.268 | 0.001 | 16.89 | ±0.558 | <0.001 | 27.25 | ±0.859 | <0.001 | 1,332 |
| Induction time | 0.38 | ±0.051 | 0.518 | 2.27 | ±0.080 | <0.001 | 4.87 | ±0.167 | <0.001 | 6.80 | ±0.258 | <0.001 | 1,331 |
| Operation time | 2.03 | ±0.136 | 0.196 | 4.40 | ±0.215 | 0.016 | 11.15 | ±0.448 | <0.001 | 15.53 | ±0.686 | <0.001 | 1,328 |
| Emergence time | −0.08 | ±0.031 | 0.823 | 0.04 | ±0.046 | 0.914 | 0.33 | ±0.099 | 0.534 | 1.65 | ±0.151 | 0.011 | 1,332 |
| Recovery time | −9.28 | ±0.587 | 0.171 | −16.23 | ±0.950 | 0.044 | −0.17 | ±1.567 | 0.987 | −12.59 | ±3.351 | 0.382 | 764 |
| Total hospital stay (days) | 0.00 | ±0.00 | 0.998 | −0.03 | ±0.011 | 0.780 | 0.18 | ±0.023 | 0.148 | 0.76 | ±0.036 | <0.001 | 1,332 |
| Total number | 512 | | | 275 | | | 113 | | | 70 | | | 1,332 |

**Table 4 Unadjusted difference.** Difference in mean times when compared to normal BMI patients (BMI 18.5–24.9), unadjusted.

| BMI range | 25–29.9 | | | 30–34.9 | | | 35–39.9 | | | 40–50 | | | Total number |
|---|---|---|---|---|---|---|---|---|---|---|---|---|---|
| | Difference | 95% CI | p | Difference | SD | p | Difference | SD | p | Difference | SD | p | |
| Total time in OR | 2.794 | ±0.169 | 0.153 | 7.435 | ±0.269 | 0.001 | 17.460 | ±0.565 | <0.001 | 28.980 | ±0.869 | <0.001 | 1,332 |
| Induction time | 0.412 | ±0.050 | 0.475 | 2.286 | ±0.079 | <0.001 | 4.886 | ±0.167 | <0.001 | 6.825 | ±0.247 | <0.001 | 1,331 |
| Operation time | 1.247 | ±0.135 | 0.425 | 4.163 | ±0.215 | 0.022 | 11.530 | ±0.455 | <0.001 | 16.940 | ±0.695 | <0.001 | 1,328 |
| Emergence time | −0.339 | ±0.133 | 0.921 | 0.108 | ±0.047 | 0.784 | 0.400 | ±0.098 | 0.534 | 1.761 | ±0.151 | 0.010 | 1,332 |
| Recovery time | −9.984 | ±0.580 | 0.136 | −17.100 | ±0.936 | 0.031 | 0.077 | ±1.416 | 0.994 | −10.270 | ±3.341 | 0.474 | 764 |
| Total hospital stay (days) | −0.023 | ±0.007 | 0.773 | −0.057 | ±0.011 | 0.550 | 0.167 | ±0.024 | 0.189 | 0.753 | ±0.036 | <0.001 | 1,332 |
| Number of patients | 512 | | | 275 | | | 113 | | | 70 | | | 1,332 |

**Table 5 Controls: difference in time points holding all other control variables constant.** The Sex column is the difference in mean time between males and females with all other variables held constant. The Age column is the difference in mean time between an individual and another 1 year younger with all other variables held constant. The Height column is the difference in mean time between for an individual and another 1 inch shorter with all other variables held constant. The ASA column is the difference in mean time between ASA 3 and ASA 2 with all other variable held constant.

| BMI range | Sex | | | Age | | | Height | | | ASA | | | Total number |
|---|---|---|---|---|---|---|---|---|---|---|---|---|---|
| | Difference | SD | p | Difference | SD | p | Difference | SD | p | Difference | SD | p | |
| Total time in OR | 1.072 | ±0.186 | 0.620 | −0.372 | ±0.008 | <0.001 | −22.730 | ±1.862 | 0.024 | 3.430 | ±0.362 | 0.026 | 1,332 |
| Induction time | 0.374 | ±0.056 | 0.564 | 0.009 | ±0.002 | 0.631 | −2.483 | ±0.558 | 0.412 | 0.490 | ±0.108 | 0.289 | 1,331 |
| Operation time | 0.383 | ±0.151 | 0.824 | −0.328 | ±0.006 | <0.001 | −20.520 | ±1.484 | 0.011 | 1.794 | ±0.288 | 0.146 | 1,328 |
| Emergence time | 0.443 | ±0.033 | 0.245 | −0.013 | ±0.001 | 0.247 | −2.067 | ±0.329 | 0.245 | 0.498 | ±0.064 | 0.067 | 1,332 |
| Recovery time | 2.490 | ±0.674 | 0.748 | 0.255 | ±0.026 | 0.748 | −68.030 | ±6.780 | 0.065 | 8.572 | ±1.287 | 0.118 | 764 |
| Total hospital stay (days) | −0.012 | ±0.008 | 0.895 | 0.005 | ±0.000 | 0.039 | −0.925 | ±0.0779 | 0.029 | 0.183 | ±0.015 | 0.005 | 1,332 |
| Total number | 512 | | | 275 | | | 113 | | | 0 | | | 1,332 |

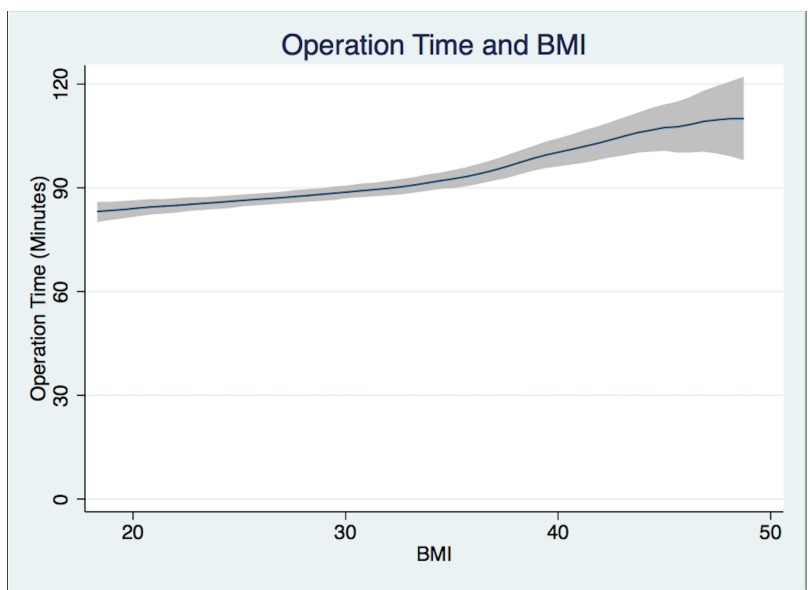

**Figure 3 Operation time.** Operation time (from surgery start time to surgery end time in minutes) and BMI Polynomial Regression Line (blue line) with 95% CI (gray); (Epanechnikov kernel of degree 0, Bandwidth of 3.02, pwidth of 4.53). The differences in the bandwidth are due to differences in the variance of outcome variables.

The greatest single contributor to the increased perioperative time requirements was the time needed by the surgeon to actually perform the operation (Operation Time). Based on this information our facility has begun a pilot project to schedule surgical time duration for THA cases by utilizing the difference in mean time between patient's BMI using Tables 3–5, which provides the absolute difference in mean time for each time interval when compared to a normal weight patient and controls.

The reasons increasing weight lengthens both anesthesia and surgical times are multifactorial but could be related to difficulties transferring patients to the OR table, with airway management, with moving the patient to the lateral decubitus position once anesthetized, and because excessive fat can interfere with intraoperative surgical exposure (*Raphael et al., 2013*). Obesity was not associated with longer recovery time, probably because discharge from the recovery room is often dictated by non-medical issues such as waiting for personnel to transport the patient to the ward or the availability of a ward bed.

It is estimated that by 2030, the number of primary THA operations will rise by 174%, and a significant number of these patients will be obese (*Kurtz et al., 2007*). Health systems will have to absorb the additional financial burden of performing THA procedures on obese patients. Those private orthopedic practices with a focus on maintaining a predictable volume and minimizing costly complications may become reticent in accepting obese patients. Besides the economic impact, the medical co-morbidities associated with obesity pose management challenges for both surgeons and anesthesiologists. With recent protocols penalizing health care providers for hospital readmissions and postoperative complications, private practitioners may be less incentivized to perform THA procedures on the obese. These factors all underscore that public payer health systems may be required

to absorb much of the financial repercussions of increased resource utilization for obese surgical patients.

Throughout the US there is wide variability reported for actual operating room costs depending on whether fixed costs or variable costs are included and on the specific resources consumed by each patient (*Shippert, 2005*; *Macario, 2010*). Besides the increased operating room time costs attributable to obesity, our study also noted expensive increased postoperative hospital length of stay. In addition, other studies have also shown that obese patients have a higher rate of prosthesis failure, hip dislocations, and wound infection (*Perka et al., 2000*; *Sadr et al., 2008*). Currently pay-for-performance policies do not take into account obesity as a cause for increased costs in the perioperative setting (*Hirose et al., 2011*; *Dowsey, Liew & Choong, 2011*). Some facilities may choose not to operate on the morbidly obese unless reimbursement is changed, especially if hospitals are penalized for readmissions under changing health policy regulations. We believe it would be worthwhile to evaluate the usefulness of a comprehensive protocol approach to the care of the morbidly obese surgical patient to ensure that appropriate pre-habilitation, proper equipment, experienced nurses, and other resources are allocated to these patients.

In our analysis, the mean operation and induction times plateau and begin to decrease as the extremes of BMI are reached. The confidence interval widens as the BMI increases, which is a result of a limited number of morbidly obese patients undergoing hip arthroplasty as compared to normal BMI, overweight, and obese patients undergoing this operation.

This study has inherent limitations in the fact that it is a single-center retrospective electronic chart-review. The time points collected by preoperative, OR, and recovery room nurses may have some errors, but should affect the BMI groups similarly. In the case of recovery times, a large number of the entered values were not recorded by the STRIDE database properly limiting the power of our recovery time results. The multiple $t$-test analysis is an inherent limitation in that we can only view each $t$-test alone, since there is inherent correlation between time intervals. We decided to report the most straightforward and easily understood test in the presentation of the data. Despite this, it is apparent from the data presented that the higher the BMI the longer the OR time will be. This will hold true however the data is reported. An inherent limitation in retrospective non-randomized studies is the fact that causal inferences can be mistakenly made, but we are confident in this study, which is supported by current available literature. As can be seen in Figs. 1 and 2, the case time decreases slightly as the confidence interval widens. Outliers were removed as mentioned above in the statistical analysis section. At the higher end of BMI there is a paucity of data which leads to these potentially deceptive graphical representations. The majority of total hip arthroplasties at our institution are performed under general anesthesia, with or without spinal for post-operative pain control. At a majority of institutions across the country spinal anesthesia alone is the anesthetic plan of choice. Despite this we believe that operation time and other secondary outcomes could be of use.

## CONCLUSION

We found that for patients undergoing THA, increasing BMI was associated with increased total OR time. Also increasing BMI was associated with longer hospital stays after THA. Operating room scheduling and plans for resource utilization should recognize that the same THA procedure will require more time in a morbidly obese patient than in a normal-weight or pre-obese patient. These considerations can potentially reduce the increased healthcare costs associated with performing surgery on obese patients.

## ACKNOWLEDGEMENTS

We wish to acknowledge the contributions of Annie Katz, MD, Resident, Internal Medicine, Stanford University Medical Center, Stanford, CA, USA for critical analysis of the proposal and manuscript; Pietro Tebaldi, PhD, Stanford University, Stanford, CA, USA for aiding in data analysis; and John Brock-Utne, MD, Professor of Anesthesiology, Department of Anesthesiology, Perioperative and Pain Medicine, Stanford University Medical Center, Stanford, CA, USA for critical analysis of the abstract.

### Funding

No funding was obtained for this study.

### Competing Interests

The authors declare there are no competing interests.

### Author Contributions

- Bassam Kadry conceived and designed the experiments, performed the experiments, wrote the paper, reviewed drafts of the paper.
- Christopher D. Press conceived and designed the experiments, performed the experiments, wrote the paper, prepared figures and/or tables, reviewed drafts of the paper.
- Hassan Alosh conceived and designed the experiments, wrote the paper.
- Isaac M. Opper, Joe Orsini and Igor A. Popov analyzed the data.
- Jay B. Brodsky conceived and designed the experiments, wrote the paper, reviewed drafts of the paper.
- Alex Macario contributed reagents/materials/analysis tools, reviewed drafts of the paper.

### Ethics

The following information was supplied relating to ethical approvals (i.e., approving body and any reference numbers):

Stanford University School of Medicine #28653.

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
