# Peer review of "Obesity increases operating room times in patients undergoing primary hip arthroplasty: a retrospective cohort analysis"

_PeerJ, doi:10.7717/peerj.530_

## Round 0.1 · original submission · Major Revisions

· Academic Editor

Major Revisions

Dear Dr Kadry and colleagues,

Thank you for the submission of your manuscript to PeerJ.

I have carefully read your manuscript, and three reviewers with expertise in the subject area have also rendered opinions. In general, you have written a good manuscript investigating an important subject. Your paper is particularly interesting when considered in the light of some medical practices/hospitals potentially excluding obese patients for joint arthroplasty because of disincentives and penalties related to patient size (as you discuss in your paper). Therefore, your paper has significant relevance to current medical practice.

However, as interesting as your paper is, there are some methodological issues which need further clarity.

I would be grateful if you would examine the enclosed reviewers' comments, make the requested modifications, and submit a revision. Please ensure that you respond in detail to each suggestion, even those you decide not to incorporate in your revision.

In addition to the Reviewers’ points, it is also important to address the following elements of your manuscript:

- L68: why were patients having spinal anesthesia not included? Many centres perform > 90% of THAs under spinal anesthesia, which may limit the external validity of this paper’s findings. (This should definitely be discussed under the Limitations section, too).
- L107: how was the regression model constructed (i.e. how did you decided which covariates to include/exclude)? Please make this explicit in the paper.
- re: regression models, I am confused because it appears that you sometimes entered BMI as an indicator variable (see Table 3 where categories of BMI are presented) and sometimes as a continuous variable (i.e. for your polynomial regression models). Therefore more clarity about how BMI was treated, and your statistical analyses in general, is needed.
- L110: a small point, but Stata is not spelled “STATA” (see http://www.stata.com/statalist/archive/2006-11/msg00039.html if you happened to be interested in the pedantry)
- L130: change “Patient” to “Patients”
- L130 to L138: all of the point estimates and 95% CIs in this paragraph are simply recapitulated from the data already included in Table 2. Please remove and describe only using text and refer the reader to the Table
- Table 3: I’m assuming this represents the coefficients of the BMI covariate (as indicator variables) in each of the regression models. However, the best way of reporting this is also to include all other variables entered in the models so the reader can see how the unadjusted coefficients (from univariable associations) changed when multivariable regression was done. (See below under STROBE recommendations.) Please update this Table with the other covariates, their uncertainty (95% CIs) and their individual P values. It may be necessary to construct individual Tables for each of the models.
- Limitations: some mention of the limitations of regression models with respect to causal inference should be made. Residual confounding because of unknown/unmeasured confounders is always an issue with regression models, and this should be highlighted.
- It seems that the caption for Table 1 is incorrect? It says “Controls – Mean times (in minutes), associated standard deviation and p-value: heteroscedastic T-Test Results Comparing Interval Times of Normal BMI to other BMI
Grouping” but appears to show demographic variables… Please clarify/correct.
- Table 1: I agree with the Reviewer that it is unacceptable to analyze ASA status and sex as continuous variables. These variables should be re-analyzed as a contingency table (ASA by obesity grouping) using Fisher’s exact test, and presented as n/%.
- Tables: If in fact you used Stata’s t test with unequal variances (i.e. “ttest, unequal”) then please refer to these t tests as “t tests assuming unequal variances”. Heteroskedasticity is a concept that is unlikely to be familiar to most readers.
- Table 2: please include SD of times as an additional column for each BMI grouping. Although it is possible to work backwards from the 95% CIs, it is cumbersome to do so.
- where P values in Stata are reported as “0.000” it actually should be reported as “< 0.001”, or, almost equivalently, “< 0.0005”. Please update your Tables.
- Figures: please reformat these graphs so that the vertical axes start at zero. Also, although the kernel details may of some interest to some statistically-inclined readers, most will not understand it. The details of the kernel could be described in your Methods section, or in the Figures’ captions, rather than on the graph itself. (I do realize this is Stata’s default, but they can be removed.)
- Generally, as also mentioned by a Reviewer, I would encourage you to update the reporting of this paper using the STROBE guidelines (von Elm E, Altman DG, Egger M, Pocock SJ, Gøtzsche PC, Vandenbroucke JP, and STROBE Initiative. The Strengthening the Reporting of Observational Studies in Epidemiology (STROBE) statement: guidelines for reporting observational studies. J Clin Epidemiol. 2008, Apr;61(4):344-9.). Once done, this can also be mentioned in your manuscript at the start of your Methods section. You will note that one of their recommendations is “Give unadjusted estimates and, if applicable, confounder-adjusted estimates and their precision (e.g., 95% confidence interval). Make clear which confounders were adjusted for and why they were included”. In your revision, please include a Table showing, at least for the primary regression model, all of the covariates included in the model, their unadjusted coefficients and the adjusted coefficients (and their 95% CIs and P values).
- I think it is OK for you to include Figures B and C, in contrast to what one of the Reviewers stated.

Please note I can make no guarantee of acceptance after revision. Your revision will be peer-reviewed once again before a decision on publication is made. Thank you again for your submission to PeerJ.

Philip M Jones, MD MSc (Clinical Trials) FRCPC
* * *
·

Basic reporting

-relevant prior literature regarding obesity and increased costs/times in THA should be mentioned in the introduction:
1) Public Health Nutr. 2009 Aug;12(8):1122-32. doi: 10.1017/S1368980009005072. Epub 2009 Mar 12. Impact of body mass on hospital resource use in total hip arthroplasty. Batsis JA1, Naessens JM, Keegan MT, Wagie AE, Huddleston PM, Huddleston JM.
(discusses anesthesia times and OR times across BMI categories for THA in Table 2)
2) Clin Orthop Relat Res. 2014 Apr;472(4):1232-9. doi: 10.1007/s11999-013-3316-9. Epub 2013 Oct 8. Obesity increases length of stay and direct medical costs in total hip arthroplasty. Maradit Kremers H1, Visscher SL, Kremers WK, Naessens JM, Lewallen DG.
(discusses health costs across BMI categories for THA)
-Specifically, what the current study adds should be noted (particularly since anesthesia and OR times have been previously described)

Experimental design

-the study should conform to the "STROBE" guidelines
(J Clin Epidemiol. 2008 Apr;61(4):344-9. doi: 10.1016/j.jclinepi.2007.11.008.
The Strengthening the Reporting of Observational Studies in Epidemiology (STROBE) statement: guidelines for reporting observational studies. von Elm E, Altman DG, Egger M, Pocock SJ, Gøtzsche PC, Vandenbroucke JP; STROBE Initiative.)
-the study design should be stated in the title
-the research question should be more clearly defined - 'measure the relationship' and 'how obesity contributes' are somewhat vague (line 55-57). It may help to state the pre-specified hypothesis to define the research question more clearly.
-the cohort generation should be better described, including the number of cases excluded for each reason (e.g. revision arthroplasty, ASA4, regional anesthesia used, lines placed). A flow chart may help.
-if there were a large number excluded due to the use of regional anesthesia, doing a separate analysis for those undergoing regional anesthesia may improve external validity, since this is a common anesthesia technique for THA (i.e. do the results hold true for both general and regional anesthesia). If few received regional anesthesia, then excluding is acceptable. Whether there is a separate 'block room' would also need to be considered.
-specific type of regression analysis and rationale for why it was chosen should be stated early (line 103); type of regression analysis should also be stated in the abstract
-were any sensitivity analyses done to determine how robust the findings are?
-tests for trend may be more meaningful for Table 2 than individual p-values comparing to normal weight and lessens the risk of type 1 error due to multiple testing
-Table 3 does not appear to add much additional information beyond Table 2, but if it is left in place, it should include confidence intervals
-Case time and anesthesia time appear to decrease at highest BMI (Figure A&B), but not surgical time (Figure C) - this could be discussed, or at least stated that the CI is large at extreme BMI due to low number of subjects; was an analysis of potential outliers and influential points done?
-for surgical time (Figure C) - how well does a linear model compare to the local polynomial? (if a linear model or 2 linear splines are a good fit, it may provide a simpler interpretation)

Validity of the findings

-are the data available per the PeerJ policy "The data on which the conclusions are based must be provided or made available in an acceptable discipline-specific repository"?
-less emphasis should be given to cost-reduction in the discussion, given that only OR times and not costs were studied. This should at least be identified as a limitation.

Additional comments

Overall a well laid out study of an important topic.

·

Basic reporting

Figure B should read Induction Time, not Anesthesia Time, to be consistent with text

Figure C should read Operation Time, not Operating Time, to be consistent with text

Experimental design

No Comments

Validity of the findings

No Comments

Additional comments

It would be interesting to see the study results including spinal as the anesthetic.

Reviewer 3 ·

Basic reporting

1. Interesting paper that reinforces what one would expect, but quantifying the effect seems valuable.
2. Clarify what happens between times (c) to (d). Why would this be expected to different with differing BMI patients.
3. Grammar line 98.
4. The results are interesting, but the report should concentrate on the important aspects. Delete Figures B & C which add nothing to the report. The text explanation that the majority of the observed difference is related to the surgical time is sufficient.
Delete Table 3 and remove "using Table 3" from line 150
5. The limitations section 187-189 should could be written more smoothly.

Experimental design

1. Need to clarify why ASA 1, 4, and 5 patients excluded. One wonders if the analysis was performed on all of the groups and the most promising results selected. Also need to explain chosen date range - it is not an even number of years. It makes one consider data mining.
2. I am not a statistician; this manuscript should be reviewed by one.
3. Table 1 shows mean +/- SD for ASA status, which is a non-parametric variable. I don't think mean +/- SD is the correct way to present ASA PS data and I wouldn't think a t-test is the appropriate comparison.
4. Gender is shown as male 0.32 +/- 0.47 - what does this mean? How does one apply a t-test to gender data?
5. Is it correct to sequentially apply a t-test on successive data sets?

Validity of the findings

Figure A. The curve reaches a maximum at BMI 46 and then declines again, which is quite surprising to me. should be explained/commented.

Why are there no patients with BMI > 50, were they excluded by the authors, by the institution?

Additional comments

It is interesting that the BMI categories had similar age characteristics, I would not have expected that. Is it worth commenting on?

---

## Round 0.2 · Minor Revisions

· Academic Editor

Minor Revisions

Dear Dr Kadry and colleagues,

Thank you for submitting your revised manuscript. This paper is now provisionally accepted for publication, but there are still several important changes which need to be incorporated into a further revision before your paper will be acceptable for publication.

First, my thoughts on the Reviewers’ comments:

Susan Lee
* * *
I agree with Dr Lee’s comments. Note especially that the statistical reporting of this manuscript still needs substantial work to be acceptable. Please leave Table 2 as-is, using SD as the estimate of variability of the data. However, in Tables 3, 4, and 5, please note plausible ranges of the differences and of the regression coefficients by using the 95% CI (rather than SD). Additionally, clarity is needed in describing the values presented. In Tables 3 and 4, “control” seems to refer to adjusted. If true, please report the unadjusted Table first, and then the adjusted Table rather than the other way around. Use the terms “unadjusted” and “adjusted” rather than “controls” or “without controls”.

Other Reviewer
* * *
Please ignore the instruction to delete Tables 3-5 as well as Figures 2/3.
I have assessed the statistical validity of your work. There will not be an outside review of this.

Other Comments:
==========
- please ensure the system accepts your new title, as Page 1 of the Reviewing PDF still has the old one
- L66: perhaps rather than mentioning type II error, an alternative would be something like “because there were very few of these patients”
- L70: > 40 should be >= 40
- L103: you already mentioned the CVP/art line issue on L74. Please delete this one.
- L122: BMI is “an” indicator variable, not “the”
- L212: suggest changing “deceiving” to “potentially deceptive”
- L217/218 have been repeated (L207/208). Remove one of them.
- caption for Table 1 has repeating text
- Table 5: it is crucial for the interpretation of this Table that the caption of the Table explains what the incremental unit is for each regression parameter. Example, it must be clear what “Sex” is - going from female to male or vice-versa? Likewise for each parameter. An example might be something like “For every year increase in age, total time in OR is reduced by 0.37 minutes.”. If this is not done, ambiguity is the result. For instance, for height, is the increment inches, cm, or feet?
- Figures: It sounds like you do not understand the method of smoothing well if you state “ default options that Stata chose to optimize the figures”. Please omit or change.

Please revise and resubmit your manuscript. Thank you.

Philip M Jones, MD, MSc (Clinical Trials), FRCPC
University of Western Ontario

·

Basic reporting

-thank-you for incorporating the retrospective analysis in the title, however it should report what type of retrospective analysis it is (e.g. cohort)

-the statement 'each incremental BMI unit increase was associated with greater incremental total OR time increases' in the abstract should be quantified (in fact, I can't really figure out where this came from in the results section since BMI was never treated as a continuous variable to produce a result per unit change, unless you are referring to your unadjusted graphical model?)
-line 42 - probably enough to say it is relevant, 'very' relevant doesn't add much, unless you explain in further detail to whom it is especially relevant (e.g. managers, clinicans, etc.)

Results section:
-indicate that the times you are reporting are means
-back-up your statements with appropriate statistical testing (e.g. line 136-137 - is this based on the various t-tests or a test for trend? do you mean BMI category, since incremental unit changes in BMI were not assessed)
-tables 3-5 should be presented in the results section. It is unusual for the first reference to a table to appear in the discussion.

Comments on Tables:
-it is a lot of work for a reader to calculate the 95% CI from the SD, while Table 2 is appropriately reported as mean +/- SD, Table 3 should be reported as 95% CI of the difference, which would give the reader a range of values consistent with what was found in the study

-'ns' p-values should still be reported. (a p-value of 0.051 is quite different than 0.9)
-p<0.05 should be reported as an actual number (e.g. p=0.04); the inequality should be reserved for only very small p-values (e.g. p<0.001)

-why are so many patients missing recovery times? (only 764 vs. 1332 for other times)

-the language 'with controls' and 'without controls' for tables 3-5 is confusing. Is what is meant actually 'unadjusted' and 'adjusted'? In which case, it should be clearly re-stated in the footnote that these values are adjusted for sex, age, etc.
-Dr. Jones' comment #7 suggestion to make each model into a table might make the interpretation clearer to the reader

Experimental design

line 66 -it is unclear to me why removing ASA 1, 4, and 5 patients would reduce type II error - please explain

thank-you for the text explanation of excluded patients, however, I think it would make things clearer to the reader to report on the actual number of patients excluded at each stage (either in text or as a flow chart). This type of transparency allows the reader to judge for themselves how likely selection bias is occurring. Without knowing how many were excluded, it limits the external generalizability and may introduce bias.

-I understand that your institution does not routinely perform spinal only, but it would be good to know the number of patients that received spinal for postop pain control as you mention they are included - does the spinal occur in the OR or elsewhere? If in the OR, this could be a potential reason for increased induction time so it is important to let the reader know and also consider adjusting for it in the multivariable regression. Also, instead of dismissing the possibility of a decrease in induction time at extreme BMI, perhaps these are the patients where nobody attempts spinal for postop pain control and therefore speeds up induction?

Validity of the findings

line 194-195 - this contradicts the statement in the abstract ("at a BMI>35 each incremental BMI unit increase...")

-I would still encourage sensitivity analyses in specific subgroups (e.g. those that did not receive spinal for postop pain, inclusion of ASA 4/5, inclusion of BMI >50, etc.) to test the robustness of the results; also, since surgeon and anesthesiologist data are available and mentioned, a clustered analysis accounting for these would be appropriate since it is reasonable to assume that average surgical times could differ between surgeons / induction times differ between anesthesiologists

Additional comments

With a little more clarity of the methods and statistical reporting, this will be an interesting read.

Reviewer 3 ·

Basic reporting

1. Tables 1 and 2 contain the relevant data. Tables 3-5 should be deleted or published as supplementary data.
2. Figures 2 and 3 add nothing to the manuscript and should be deleted. Text description is likely adequate.
3. Define: comprehensive "service-line"
4. Tables should round % data to nearest percent.

Experimental design

1. I still have grave concerns about the statistical validity of this analysis, comparing arbitrary groups to "normal" successively with a t test, instead of performing a multivariate analysis. I will defer to the PeerJ statistician on this matter.
2. Need to clarify number of removed data points (called "few" in manuscript)
3. "since there is inherent correlation between these time intervals" This makes no sense.

Validity of the findings

1. Limitations section contains many inaccuracies.
2. "We decided to report the most straightforward and and easily understood test" - a better idea would be to report the correct test.
3. Grammar: Despite this.. "will be" should be "was"
4. The limitation of mistaken causal inference is related to the retrospective nature and lack of randomization, not regression modelling, and this study ALSO suffers from this potential malady.
5. I don't believe the graphical representations are "deceiving" at all.
6. The last sentence of the limitations section repeats a previous sentence.

Additional comments

Grammar: in ability, we reduce, Also increasing,
"All were the default options that Stata chose to optimize the figures." sounds odd.

---

## Round 0.3 · accepted · Accept

· Academic Editor

Accept

Thank you for adequately responding to my comments and to those of the Reviewers. I am pleased to let you know that your paper is now accepted for publication in PeerJ.